# ARE CLASSIFICATION ROBUSTNESS AND EXPLANATION ROBUSTNESS REALLY STRONGLY CORRELATED? AN ANALYSIS THROUGH INPUT LOSS LANDSCAPE

## ABSTRACT

This paper looks into the critical area of deep learning robustness and challenges the common belief that classification robustness and explanation robustness in image classification systems are inherently correlated. Through a novel evaluation approach leveraging clustering for efficient assessment of explanation robustness, we demonstrate that enhancing explanation robustness does not necessarily flatten the input loss landscape with respect to explanation loss - contrary to flattened loss landscapes indicating better classification robustness. To further investigate this contradiction, a training method designed to adjust the loss landscape with respect to explanation loss is proposed. Through the new training method, we uncover that although such adjustments can impact the robustness of explanations, they do not have an influence on the robustness of classification. These findings not only challenge the previous assumption of a strong correlation between the two forms of robustness but also pave new pathways for understanding the relationship between loss landscape and explanation loss. Codes are provided in the supplement.

## 1 INTRODUCTION

In deep learning, the robustness of image classification systems against adversarial instances has emerged as an important area of research. These systems, integral to modern artificial intelligence, frequently encounter scenarios where adversarial instances—subtly altered images designed to deceive algorithms—pose significant challenges. At the heart of this challenge lie two critical concepts: classification robustness and explanation robustness. Classification robustness refers to a model's ability to maintain accuracy under adversarial attacks (Szegedy et al., 2013; Madry et al., 2017), while explanation robustness pertains to the consistency of the model's interpretative outputs in such adversarial scenarios (Dombrowski et al., 2019; Huang et al., 2023). Traditionally, there's been a prevailing conclusion within the research community (Boopathy et al., 2020; Huang et al., 2023):

*Conclusion: Classification robustness and explanation robustness are strongly correlated:*
*Increasing classification robustness can increase explanation robustness and vice versa.*

This paper, however, unveils a finding that disrupts this conventional belief: a contradiction in the assumed correlation between classification robustness and explanation robustness. This revelation not only challenges established assumptions but also opens new avenues for understanding and improving the resilience of deep learning models.

Adversarial attacks on classification aim at deceiving image classification models by introducing perturbations to benign images (Szegedy et al., 2013). To defend against adversarial examples, adversarial training (AT) (Madry et al., 2017; Goodfellow et al., 2014) is one of the most effective approaches which explicitly augments the training process to enhance a model's inherent robustness against adversarial samples for classification. Classification robustness typically is referred as the classification accuracy under adversarial attacks, and AT methods are effective in improving the classification robustness of a deep learning model.

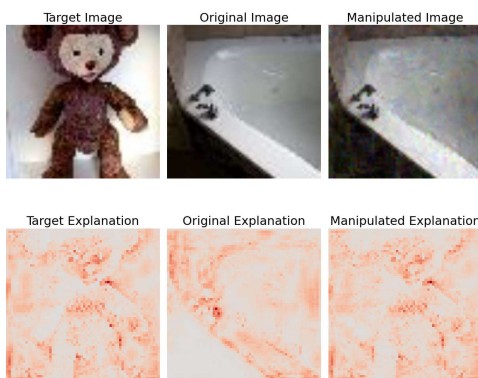

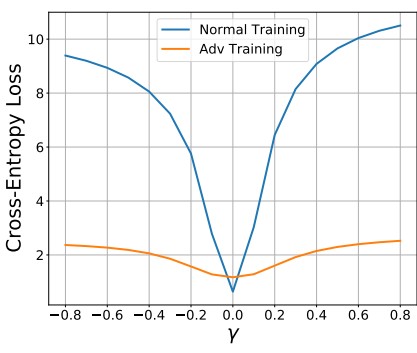

(b) Comparison of input loss landscapes between normal and adversarially trained models on CIFAR-10 shows that the adversarially trained models have much flatter landscapes. Adversarial training can increase classification robustness.

(a) Example of adversarial attack on explanation. The explanation maps of original image can be manipulated to the target explanation.

Figure 1: (a) Illustration of an adversarial attack on explanation, demonstrating the manipulation of explanation maps from the original image to achieve a target, resulting in explanation loss (b) A visualization of input loss landscape w.r.t classification loss, comparing a normal-trained model to an adversarial-trained model.

Explanation maps (Baehrens et al., 2010), also known as saliency maps, are proposed to explain deep learning methods by feature importance. However, explanation maps are themselves also vulnerable to adversarial attacks (Dombrowski et al., 2019; Ghorbani et al., 2019). For example, in Fig. 1a, by making small visual changes to the input sample which hardly influences the network's output, the explanations can be arbitrarily manipulated (Dombrowski et al., 2019). Explanation robustness is referred as the error between victim explanation under adversarial attacks on the input and targeted explanation.

To better understand robustness, one important way is to explore the input loss landscape (Li & Spratling, 2023). Existing work has found out that *a flat input loss landscape w.r.t classification loss indicates better classification robustness* (Xie et al., 2020; Li & Spratling, 2023), as shown in Fig. 1b. To visualize the input loss landscape, we add the random perturbation to the inputs with magnitude $\alpha$ (detailed method in Section 4). The results in Fig. 1b show that models with higher classification robustness have a flatter input loss landscape w.r.t classification loss.

Then a natural question comes up for explanation robustness:

*Q: Does increasing explanation robustness of a model also flatten input loss landscape w.r.t explanation loss?*

We visualize the input loss landscape w.r.t explanation loss in Fig. 4 using models with different levels of explanation robustness and find that, surprisingly, *increasing the explanation robustness does not flatten the input loss landscape w.r.t explanation loss*. Specifically, to obtain models with different levels of explanation robustness, we consider utilizing adversarial training methods that allow us to control the emphasis on classification robustness (Zhang et al., 2019) since previous works have proven that increasing classification robustness can also increase explanation robustness.

The previous observation that increasing the explanation robustness does not flatten the input loss landscape w.r.t explanation loss is strange compared with increasing classification robustness could flatten the input loss landscape w.r.t classification loss. To further explore this observation, we ask the previous question in a reverse way:

*Q: Does flattening the input loss landscape w.r.t explanation loss not increase the robustness of explanations as well?*

The answer to this question is, flattening the input loss landscape w.r.t explanation loss will decrease the explanation robustness. Specifically, we propose a new loss function to flatten the loss landscape w.r.t explanation loss. The results show that adding the loss will decrease the explanation robustness

but not change the classification robustness measured by adversarial accuracy. This observation, indicating that influencing explanation robustness does not impact classification robustness, challenges the previous conclusion: the correlation between explanation robustness and classification robustness may not hold.

Overall, we summarize our contributions as follows:

- We propose a sampling method based on cluster methods that can choose representative pairs to evaluate explanation robustness more efficiently.
- We use TRADES (Zhang et al., 2019) to control the classification robustness and explanation robustness and visualize the input loss landscape w.r.t explanation loss to find that increasing the explanation robustness by increasing the classification robustness does not flatten the input loss landscape.
- We propose a new training method that flattens the input loss landscape w.r.t explanation loss. The training results show that explanation robustness may **not** be strongly correlated to classification robustness.

## 2 RELATED WORK

**Adversarial Attack and Adversarial Training (AT)** It has been proven that convolutional neural networks (CNNs) are vulnerable to the adversarial examples (Szegedy et al., 2013; Goodfellow et al., 2014; Carlini & Wagner, 2017). Noise that is imperceptible to humans, when added to the original inputs, can lead to the misclassification of models. Projected Gradient Descent (PGD) (Madry et al., 2017) is one of the most popular methods that generate such a noise or evaluate models' classification robustness by calculating accuracy under its attack. Many methods have been introduced to defend against adversarial attacks including knowledge distillation (Papernot et al., 2016), quantization (Xu et al., 2017; Lin et al., 2019) and noise purification (Song et al., 2017; Carlini et al., 2022). However, these preprocessing methods do not involve a training process and may be vulnerable to adaptive attack (Athalye et al., 2018). Goodfellow et al. (Goodfellow et al., 2014) first introduced adversarial training (AT), which trains a model from scratch with adversarial samples. Adversarial Training (AT) proved its performance including adversarial competitions (Madry et al., 2017; Brendel et al., 2020). In our paper, we also focus on classification robustness increased by AT.

Many works tend to increase the performance of AT through external datasets (Hendrycks et al., 2019; Carmon et al., 2019; Wang et al., 2023), metric learning (Pang et al., 2019), self-supervised learning (Chen et al., 2020a), ensemble learning (Tramèr et al., 2017), label smoothing (Chen et al., 2020b) and Taylor Expansion (Jin et al., 2023). Wu et al. (Wu et al., 2020) found that obtaining a flat loss landscape can help increase classification robustness, which inspired the ideas in this paper. There is also a line of work that attempts to accelerate AT. For example, Shafahi et al. (Shafahi et al., 2019) reused calculated adversarial noises, Liu et al. (Liu et al., 2021) introduced single-step training. In this paper, we mainly consider the Madry adversarial training (Madry et al., 2017) and TRADES (Zhang et al., 2019).

**Explanation Robustness** Saliency maps (Simonyan et al., 2013; Shrikumar et al., 2017; Bach et al., 2015; Selvaraju et al., 2016) are widely used to explain image-related tasks in deep learning, and our focus is on the robustness of these explanations. However, similar to an adversarial attack, it is possible to find an adversarial noise on original images so that it can easily manipulate the saliency maps without changing classification results in both white-box (Dombrowski et al., 2019; Ghorbani et al., 2019; Heo et al., 2019; Slack et al., 2020) and black-box settings (Tamam et al., 2022). Zhang et al. (Zhang et al., 2020) further introduced a new method that can attack both saliency maps and classification results. In order to evaluate the explanation robustness, Wicker et al. (Wicker et al., 2022) introduced the max-sensitivity and average-sensitivity of saliency maps. Alvarez et al. (Alvarez-Melis & Jaakkola, 2018) estimated explanation robustness by the Local Lipschitz of interpretation while Tamam et al. (Tamam et al., 2022) directly used attack loss to evaluate explanation robustness. In this paper, we use attack loss based on the proposed cluster method to evaluate explanation robustness.

Several works have also aimed to improve explanation robustness. Chen et al. (Chen et al., 2019) introduced a regularization term during training to make the explanation more robust. Boopathy et al. (Boopathy et al., 2020) improved the performance by training with noisy labels. Tang et al. (Tang

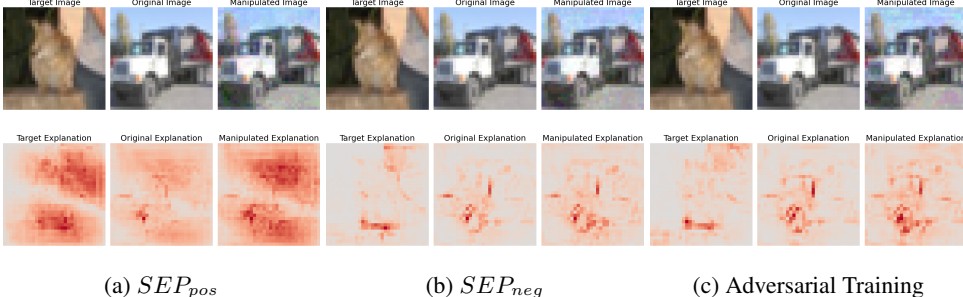

(a) $SEP_{pos}$        (b) $SEP_{neg}$        (c) Adversarial Training

Figure 2: How does our method influence the saliency maps calculated from gradient x inputs on CIFAR10. Intuitively, $SEP_{pos}$ makes the model consider more input pixels, solely adversarial training makes the model consider only a few input pixels while $SEP_{neg}$ considers even fewer input pixels compared with adversarial training. However, models trained with these three methods show the same classification robustness.

et al., 2022) proposed a first-order gradient-based approach to reduce computational training costs. Huang et al. (Huang et al., 2023) explored genetic algorithms to optimize explanation robustness.

**Relationship between Classification Robustness and Explanation** Previous works have demonstrated that a good explanation is crucial for classification robustness (Simonyan et al., 2019; Wang et al., 2021), suggesting that a better saliency map correlates with improved classification robustness. Follow-up work by Boopathy et al. (Boopathy et al., 2020), Tang et al. (Tang et al., 2022) and Huang et al. (Huang et al., 2023) further demonstrated when models are more robust to attacks manipulating explanations, their robustness to classification attacks also improves and vice versa. Therefore, increasing explanation robustness can benefit classification robustness.

In our paper, we prove that improving explanation robustness indeed also boosts classification robustness, specifically under adversarial training regimes using TRADES (Zhang et al., 2019). However, through further analysis, we prove these two facets of robustness are not inherently the same - they can be disconnected. Classification robustness is not fundamentally vital for explanation robustness.

## 3 METHODS

In the previous section, we observe a strange situation where increasing the explanation robustness does not flatten the input loss landscape w.r.t explanation robustness. To further explore, we consider this situation in a reverse way: *How Does flattening the input loss landscape w.r.t explanation loss influence the robustness of explanations?* In this section, we propose a new training algorithm to flatten the input loss landscape w.r.t explanation robustness. To explicitly guide the training with flattening input loss landscape w.r.t explanation robustness, we decide to add an extra loss:

$$\mathcal{L}_f = \|I(x + \zeta) - I(x)\|, \tag{1}$$

where $\zeta$ is a noise randomly sampled from a standard Gaussian distribution and $I$ is the explanation method. We use randomly sampled noise within a standard training framework instead of the min-max training framework used in the previous flatness-aware methods (Wu et al., 2020) because flat training methods based on AT (Wu et al., 2020) typically use an untargeted setting while off-the-shelf explanation adversarial attacks must be executed in a target setting. A victim image and a target image are required for the explanation of adversarial attacks (Tamam et al., 2022; Dombrowski et al., 2019). Besides, calculating $\zeta$ through a targeted setting may increase the training time and increase the probability that the model is overfitting to the chosen pairs.

It is important to note that the new loss function $\mathcal{L}_f$ can be incorporated into any training framework, including Madry adversarial training (Madry et al., 2017), TRADES (Zhang et al., 2019), and normal training. We will mainly focus on Madry adversarial training plus the new training loss:

$$\mathcal{L} = \mathcal{L}_{sc}(f(x_{adv}), y) + \lambda \mathcal{L}_f. \tag{2}$$

In Eq. (2), we use the hyperparameter $\lambda$ to balance two components of the loss. Here $\lambda$ can be both positive which guides the loss landscape to become flat and negative which leads to a sharper loss landscape. We allow $\lambda$ to take both positive and negative values to enable a more comprehensive analysis of the loss landscape. According to the experimental results, our new method shows that our method can influence explanation robustness while it does not change classification robustness. Since we obtain $x_{adv}$ based on PGD, we name our new training method with **S**eparate **E**xplanation robustness with **P**GD (SEP). We denote the

**Algorithm 1** **S**eparate **E**xplanation Robustness with **P**GD (`SEP`)

---
1: **Input:** Dataset $\mathcal{D}$, total training iteration $T$, explanation method $I$, model weights $\mathbf{w}$, and balancing factor $\lambda$.
2: **for** $t = 0$ **to** $T - 1$ **do**
3:     **for** batch $x$ in $\mathcal{D}$ **do**
4:         Sample a random noise $\zeta$ from a standard Gaussian distribution.
5:         Get adversarial samples (on classification): $x_{adv} = PGD(x, y)$.
6:         Calculate loss function with Eq. (2).
7:         Update $\mathbf{w} \leftarrow \mathbf{w} - \eta \nabla \mathcal{L}(f(x), f(x_{adv}), y | \mathbf{w})$
8:     **end for**
9: **end for**

---

method as $\text{SEP}_{pos}$ when $\lambda$ is positive, and as $\text{SEP}_{neg}$ when $\lambda$ is negative. We summarize our algorithm in Algorithm 1. We also visualize the comparison of saliency maps from models trained with different algorithms to provide how our methods influence the saliency maps in Fig. 2.

## 4 LOSS LANDSCAPE VISUALIZATION

In this section, we discuss our strategy for obtaining models with varying levels of explanation robustness and detail our method for visualizing the input loss landscape. Research has shown a correlation between increased classification robustness and enhanced explanation robustness (Huang et al., 2023). To achieve models with different classification robustness levels, we use TRADES (Zhang et al., 2019), which offers detailed control over classification robustness compared to methods like Madry adversarial training (Madry et al., 2017).

**Background of TRADES** TRADES (Zhang et al., 2019) is an adversarial training technique that balances classification and adversarial robustness using the loss function:

$$L_{Tra} = L_{sc}(f(x), y) + \alpha L_{adv}(f(x), f(x_{adv})), \tag{3}$$

where $f(x)$ is the model output, $L_{sc}$ is the standard classification loss, $x_{adv}$ is an adversarial example, and $L_{adv}$ computes the KL divergence between original and adversarial representations. The parameter $\alpha$ controls the importance of classification robustness, allowing precise regulation of robustness levels.

**Explanation Loss** The objective, used to guide adversarial attacks on explanations, is defined to find a small noise $\epsilon$ as:

$$\epsilon = \arg\min_{\epsilon} \|I(x_v + \epsilon) - I(x_t)\|, \tag{4}$$

where $I$ represents the explanation method, $x_t$ are target images, and $x_v$ are victim images. We formally define explanation loss as follows.

**Definition 1** (Explanation Loss). $\mathcal{L}_e(x_v + \epsilon, x_t) = \|I(x_v + \epsilon) - I(x_t)\|$.

To prevent $\epsilon$ from being too large, additional classification loss is used to ensure manipulated images yield the same classification results (Dombrowski et al., 2019; Tamam et al., 2022).

**Explanation Robustness Evaluation** To measure explanation robustness, we propose using a representative subset of test images, chosen via clustering. Clustering aims that intra-cluster pairs share similar explanations. We cluster images based on

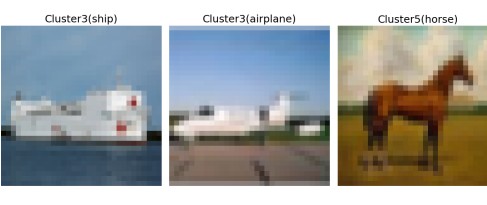
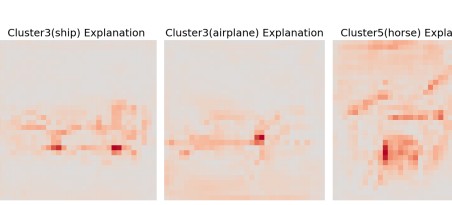

Figure 3: The explanations from different clusters generated by our clustering method on CIFAR10. The two images with different labels in the same cluster share a similar explanation while they both show a different explanation with the image from another cluster. The results show that our method can choose the most representative images w.r.t explanation.

the output from the last layer before the classification layer of a pre-trained ResNet18 (He et al., 2016) using k-means (Lloyd, 1982) with $k = 10$ by the guidance of elbow method. Visualizations of saliency maps show that images from the same cluster have similar explanations (see Fig. 3). We also report the explanation loss for intra-cluster and inter-cluster pairs to show that our clustering method indeed makes intra-cluster pairs share similar explanations quantitatively in Appendix C.

We select 15 images from each cluster to form a subset $\mathcal{D}_e$ of the test set, containing 150 images and 22,350 ($150 \times 149$) pairs. We report the mean explanation loss for all pairs in $\mathcal{D}_e$ using a white-box attack method (Dombrowski et al., 2019) to evaluate explanation robustness in the rest of the paper. We also report the explanation loss at attack starts (Expl at Start) and loss at attack ends (Expl at End) to provide a comprehensive analysis of both robustness and flatness. A higher explanation loss indicates better explanation robustness because the model is harder to attack.

Table 1: Comparison of classification robustness and explanation robustness of models trained with TRADES and different $\alpha$ on CIFAR10. Within a certain range, using the TRADES training method and increasing the value of $\alpha$ can not only improve the classification robustness but also improve the explanation robustness.

| Metric | Expl at Start($1e-7$) | Expl at End | Clean Acc(%) | Adv Acc(%) |
|---|---|---|---|---|
| $\alpha$ | Explanation Robustness | | Classification Robustness | |
| 0 | 10.375 | 6.206 | 79.08 | 0.00 |
| 0.5 | 16.635 | 10.640 | 75.60 | 23.57 |
| 1.0 | 17.271 | 10.946 | 72.63 | 28.31 |
| 2.0 | 17.290 | 10.965 | 69.56 | 31.77 |
| 4.0 | 18.004 | 11.293 | 65.63 | 33.28 |
| 5.0 | 18.278 | 11.469 | 64.50 | 33.98 |
| 10.0 | 18.643 | 11.592 | 60.26 | 34.87 |

**Analysis** In Table 1, we provide an evaluation of classification robustness and explanation robustness for models trained with TRADES and different $\alpha$ on CIFAR10. From Table 1, it is easy to see that, with the increase of $\alpha$, both the classification and explanation robustness of the model increase. Therefore, we obtain models with different explanation robustness.

**Visualization** After getting the models with different explanation robustness, the next step is to visualize input loss landscape w.r.t explanation loss. We visualize the input loss landscape by plotting the change of explanation loss when we add a random noise $\mathbf{d}$ to the victim image $x_v$ with different magnitude $\gamma$:

$$f(\gamma) = \mathcal{L}_e(x_v + \gamma \mathbf{d}, x_t), \qquad (5)$$

where $\mathbf{d}$ is sampled from a standard Gaussian distribution. We provide the mean explanation loss for all pairs in the subset we build, with the results displayed in Fig. 4. We can see that the adversarially trained models have better explanation robustness because of the high initial explanation loss instead of a flat loss landscape. We also visualize compared with normal training and Madry adversarial

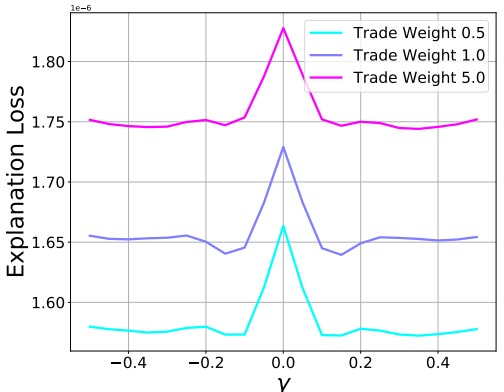

Figure 4: Input Loss landscape w.r.t explanation loss for models trained with different with different $\alpha$ in TRADES. The loss landscape does not show a clear difference between models that vary in explanation robustness because the loss change remains the same.

training (MAT) in Appendix Fig. 7, and it shows similar results: increasing explanation robustness will not flatten the input loss landscape w.r.t explanation loss. Previous work on classification robustness (Xie et al., 2020; Li & Spratling, 2023) has proven that a model with good classification robustness has a flat loss landscape w.r.t classification loss. However, different from the conclusions drawn in classification robustness, adversarially trained models don't exhibit a flat loss landscape w.r.t explanation loss. This phenomenon motivates us to propose the method in the following section to flatten the input loss landscape w.r.t explanation robustness.

## 5 EXPERIMENTAL RESULTS

In this section, we conduct verification experiments on multiple datasets and models to effectively demonstrate the ability of our proposed method to differentiate explanatory robustness from classification robustness.

Table 2: Performance of models trained with ConvNet and ResNet18 on various datasets is evaluated using four training methods, w.r.t. explanation loss at start, at end, and adversarial accuracy. Higher explanation loss at end indicates better explanation robustness; higher adversarial accuracy denotes better classification robustness. Explanation loss at start is also included to show our method's influence on explanation robustness. The **best** performance in explanation and classification robustness and the worst performance in explanation robustness are highlighted. There is no positive correlation between explanation and classification robustness achieved through $\text{SEP}_{pos}$ and $\text{SEP}_{neg}$ training methods, compared to MAT.

| | ConvNet | | | | ResNet18 | | | |
|---|---|---|---|---|---|---|---|---|
| | | | | MNIST | | | | |
| Method | Expl at Start ($10^{-7}$) | Expl at End | Clean Acc(%) | Adv Acc(%) | Expl at Start | Expl at End | Clean Acc(%) | Adv Acc (%) |
| Normal | 261.183 | 204.825 | 99.29 | 0.00 | 266.834 | 146.16 | 99.36 | 0.00 |
| MAT | 373.262 | 298.729 | 99.00 | 89.92 | 916.017 | 778.003 | 99.28 | **94.60** |
| $\text{SEP}_{pos}$ | 93.033 | 61.545 | 98.8 | 89.4 | 92.371 | 59.278 | 98.4 | 91.63 |
| $\text{SEP}_{neg}$ | 806.204 | **657.180** | 98.97 | 90.34 | 9356.306 | **8248.627** | 99.4 | 93.95 |
| | | | | FMNIST | | | | |
| Method | Expl at Start ($10^{-7}$) | Expl at End | Clean Acc(%) | Adv Acc(%) | Expl at Start | Expl at End | Clean Acc(%) | Adv Acc (%) |
| Normal | 106.530 | 72.198 | 92.32 | 0.00 | 128.640 | 69.847 | 91.57 | 0.00 |
| MAT | 386.370 | 274.267 | 62.85 | 73.98 | 588.610 | 417.031 | 79.22 | **67.10** |
| $\text{SEP}_{pos}$ | 35.588 | 22.465 | 69.88 | **86.81** | 32.466 | 22.512 | 68.75 | 56.51 |
| $\text{SEP}_{neg}$ | 1811.969 | **994.818** | 62.75 | 76.89 | 8050.942 | **7593.650** | 70.23 | 57.55 |
| | | | | CIFAR10 | | | | |
| Method | Expl at Start ($10^{-7}$) | Expl at End | Clean Acc(%) | Adv Acc(%) | Expl at Start | Expl at End | Clean Acc(%) | Adv Acc(%) |
| Normal | 10.375 | 6.206 | 79.08 | 0.00 | 13.982 | 6.130 | 81.32 | 0.00 |
| MAT | 16.913 | 6.906 | 64.85 | 35.11 | 31.959 | 21.879 | 67.22 | 29.09 |
| $\text{SEP}_{pos}$ | 3.565 | 1.269 | 64.94 | **35.25** | 11.962 | 7.958 | 66.68 | **29.69** |
| $\text{SEP}_{neg}$ | 19.002 | **7.590** | 64.56 | 34.86 | 70.159 | **36.276** | 39.17 | 29.32 |
| | | | | CIFAR100 | | | | |
| Method | Expl at Start ($10^{-7}$) | Expl at End | Clean Acc(%) | Adv Acc(%) | Expl at Start | Expl at End | Clean Acc(%) | Adv Acc(%) |
| Normal | 10.099 | 6.140 | 48.39 | 0.05 | 12.044 | 4.716 | 41.24 | 0.00 |
| MAT | 20.642 | 13.650 | 36.4 | 17.35 | 33.456 | 22.623 | 36.14 | 15.70 |
| $\text{SEP}_{pos}$ | 13.650 | 9.932 | 37.41 | **17.98** | 19.217 | 12.744 | 34.83 | 15.16 |
| $\text{SEP}_{neg}$ | 22.506 | **14.970** | 36.17 | 17.43 | 35.525 | **24.289** | 34.80 | **15.87** |
| | | | | TinyImageNet | | | | |
| Method | Expl at Start ($10^{-7}$) | Expl at End | Clean Acc(%) | Adv Acc(%) | Expl at Start | Expl at End | Clean Acc(%) | Adv Acc(%) |
| Normal | 0.966 | 0.633 | 28.71 | 0.00 | 1.131 | 0.528 | 28.34 | 0.00 |
| MAT | 2.426 | 1.728 | 25.13 | 9.55 | 3.119 | 2.349 | 26.34 | 10.81 |
| $\text{SEP}_{pos}$ | 2.242 | 1.571 | 24.83 | **9.63** | 1.967 | 1.435 | 25.96 | **10.83** |
| $\text{SEP}_{neg}$ | 3.873 | **2.610** | 24.31 | 9.61 | 4.413 | **3.016** | 26.11 | 10.74 |

## 5.1 Experimental Settings

**Datasets** To thoroughly demonstrate the impact of our proposed training method and the resulting conclusions, we conduct model training on five publicly available datasets for experiments: CIFAR10 (Krizhevsky et al., 2009), CIFAR100 (Krizhevsky et al., 2009), MNIST (LeCun et al., 1989), Fashion MNIST (Xiao et al., 2017), and TinyImageNet (Le & Yang, 2015). Their detailed descriptions can be found in Appendix A. We also consider using ImageNet (Deng et al., 2009) and the experiment results for ImageNet can be found in Appendix E.3.

**Model Architecture** In addition to utilizing diverse datasets, we have also designed four distinct models for training on these datasets, further reinforcing our conclusions. We conduct experiments on ConvNet, ResNet (He et al., 2016), Wide ResNet (Zagoruyko & Komodakis, 2016) and MoblieNetV2 (Howard et al., 2017; Sandler et al., 2018). The ConvNet model consists of three convolutional layers and one fully connected layer from Gidaris et al. (Gidaris & Komodakis, 2018). For ResNet and Wide ResNet, we use a standard ResNet18 and Wide-ResNet-28, respectively. We also adjust the ResNet, Wide ResNet, and MoblieNetV2 so that they can fit into all datasets we use. All four models employ the softplus (Zheng et al., 2015) activation function because it is better for the explanation attack method we use (Dombrowski et al., 2019).

**Explanation Methods** We mainly use: Gradient(Baehrens et al., 2010), Gradient × Input(Shrikumar et al., 2017), Guided Backpropagation(Springenberg et al., 2014),Deep Lift (Shrikumar et al., 2017) and Integrated Gradients (Sundararajan et al., 2017). We use Captum (Kokhlikyan et al., 2020) for all explanation methods.

**Training Methods** We mainly consider 2 baselines: i) normal training (Normal), ii) Madry adversarial training (MAT) (Madry et al., 2017). As mentioned in the Section 3, we explore two types of proposed method: $\text{SEP}_{pos}$ and $\text{SEP}_{neg}$. In the rest of this paper, unless specified, we will use $\lambda = 50000$ for $\text{SEP}_{pos}$ and $\lambda = -3000$ for $\text{SEP}_{neg}$.

Table 3: Performance of different explanation methods (Gradient and Guide Propagation) in the training phase is evaluated w.r.t. explanation loss at start, at end, and adversarial accuracy on CIFAR10. Higher explanation loss at end indicates better explanation robustness, while higher adversarial accuracy denotes better classification robustness. The **best** and worst performances in explanation robustness and classification robustness are highlighted. Under various explanation methods, $\mathrm{SEP}_{pos}$ shows a lower explanation loss compared to $\mathrm{SEP}_{neg}$, with similar adversarial accuracy.

| ConvNet | | | | ResNet18 | | | |
|---|---|---|---|---|---|---|---|
| Gradient | | | | | | | |
| Method | Expl at Start ($10^{-7}$) | Expl at End | Clean Acc (%) | Adv Acc (%) | Expl at Start | Expl at End | Clean Acc (%) | Adv Acc (%) |
| Normal | 7.977 | 4.591 | 79.08 | 0.00 | 11.310 | 4.671 | 81.32 | 0.00 |
| MAT | 13.810 | 8.705 | 64.85 | **35.11** | 26.899 | 18.215 | 67.22 | 29.09 |
| $\mathrm{SEP}_{pos}$ | 0.876 | 0.503 | 52.89 | 29.68 | 11.317 | 6.604 | 66.76 | **37.69** |
| $\mathrm{SEP}_{neg}$ | 13.964 | **9.290** | 53.23 | 29.56 | 8282.990 | **7236.182** | 49.38 | 32.28 |
| Guide Propagation | | | | | | | |
| Method | Expl at Start ($10^{-7}$) | Expl at End | Clean Acc (%) | Adv Acc | Expl at Start | Expl at End | Clean Acc (%) | Adv Acc (%) |
| Normal | 8.075 | 4.639 | 79.08 | 0.00 | 11.515 | 4.736 | 81.32 | 0.00 |
| MAT | 14.012 | 8.813 | 64.85 | 35.11 | 27.012 | 18.311 | 67.22 | 29.09 |
| $\mathrm{SEP}_{pos}$ | 1.023 | 0.506 | 60.27 | 33.57 | 12.004 | 7.593 | 67.16 | 30.64 |
| $\mathrm{SEP}_{neg}$ | 14.643 | **9.110** | 59.74 | **33.78** | 27.422 | **18.940** | 66.48 | **30.72** |

**Hyperparameters** For all experiments, we train our models 25 epochs with 64 as the batch size. We also consider different training epochs and our conclusion remains the same in Appendix E.5. To accelerate the training process, we use Adam (Kingma & Ba, 2014) as the optimizer. We list the detailed hyperparameters for CIFAR10 in the Appendix Table 8. We use the standard settings in adversarial training (Pang et al., 2020), with $\epsilon = 8/255$ in PGD for RGB images and $\epsilon = 0.3$ for grayscale images, and steps in PGD are set to 10 for all experiments.

**Metrics** As mentioned in Section 4, we measure explanation robustness using the explanation loss at the end (after attack). A higher explanation loss indicates a worse attack and thus better explanation robustness. We also report the explanation loss at the start (before attack) to show the influence of our method on the explanation loss landscape. For classification robustness, we report adversarial accuracy, with higher values indicating better robustness. Additionally, we include clean accuracy to ensure the models function normally in non-adversarial settings.

## 5.2 SEPARATING EXPLANATION AND CLASSIFICATION ROBUSTNESS

We conducted a series of experiments involving multiple models and datasets on Gradient × Input and results are shown in Table 2 for ConvNet and ResNet18. We have the following observations:

- On one hand, $\mathrm{SEP}_{pos}$, $\mathrm{SEP}_{neg}$, and MAT have very similar adversarial accuracy, indicating their classification robustness is similar in all datasets and models. On the other hand, $\mathrm{SEP}_{pos}$ shows the weakest explanation robustness by having the lowest explanation loss at end. Similarly, $\mathrm{SEP}_{neg}$ shows the strongest explanation robustness. These results show that *there is no inherent relationship between explanation robustness and classification robustness*. The different performance w.r.t. explanation loss at end for $\mathrm{SEP}_{pos}$ and $\mathrm{SEP}_{neg}$ is mainly induced by the difference in explanation loss at start, which is influenced by our training method by setting different $\lambda$.

- In the setting of CIFAR10 and ResNet18, increasing the explanation robustness by $\mathrm{SEP}_{neg}$ hurts the clean accuracy while it still does not change classification robustness. This observation further validates our argument: classification robustness and explanation robustness may not be strongly correlated. We provide the results for W-ResNet and MoblieNetV2 in the Appendix Table 6 and the results show a very similar conclusion to the results of ConvNet and ResNet.

## 5.3 INFLUENCE OF DIFFERENT EXPLANATION METHODS IN TRAINING PHASE

In the previous experiment, we demonstrated that our methods achieve similar classification robustness while exhibiting significantly different explanation robustness under the Gradient × Input explanation method. To further investigate whether this conclusion holds for different explanation methods, we trained models using Gradient and Guide Propagation. The results are based on CIFAR10 and are summarized in Table 5 with more datasets and more explanation methods including Deep Lift and Integrated Gradients can be found in Appendix E.4. Our observations are as follows:

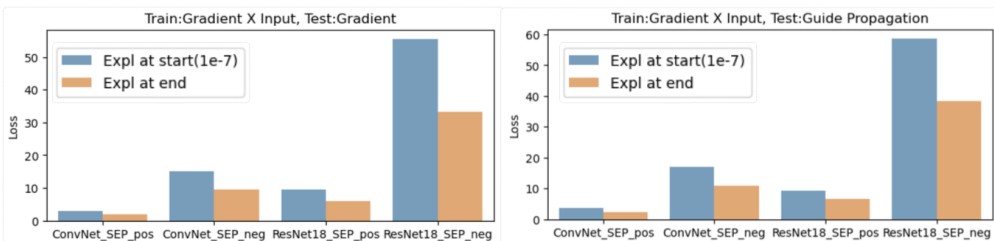

Figure 5: Performance of varying explanation methods in the testing phase, w.r.t. explanation loss at start, at end, and adversarial accuracy. Models are trained with Gradient x Input and tested on different explanation methods. All models are trained on CIFAR10. Even if the explanation methods during training and testing are different, $\text{SEP}_{pos}$ shows a lower explanation loss compared to $\text{SEP}_{neg}$, while they have similar adversarial accuracy.

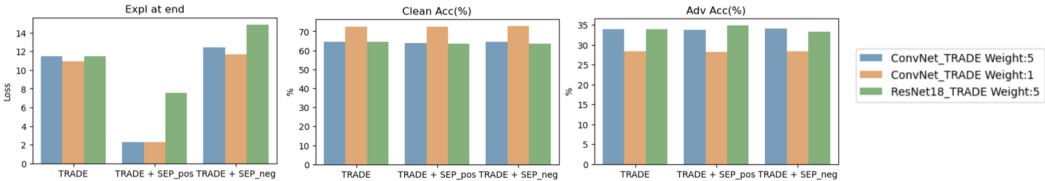

Figure 6: The test results of the model trained using the TRADE training method with CIFAR10, combined with our approach, are presented. The findings indicate that when we apply our method to TRADE, an alternative adversarial training method distinct from MAT, we can still deduce that classification robustness and explanation robustness are not inherently interconnected. This outcome demonstrates the universal applicability of our proposed method.

- Our methods achieve similar classification robustness under various explanation methods, yet they exhibit notably different explanation robustness. In most cases, $\text{SEP}_{pos}$ shows lower explanation loss compared to $\text{SEP}_{neg}$, despite similar adversarial accuracy.

- Compared to MAT, our method $\text{SEP}_{pos}$ shows comparable adversarial accuracy, indicating similar classification robustness, but it demonstrates distinct explanation loss characteristics. This suggests that explanation robustness and classification robustness may not be strongly correlated.

## 5.4 INFLUENCE OF DIFFERENT EXPLANATION METHODS IN TESTING PHASE

In the previous experiments, the same explanation methods were used during both training and testing. To test if our findings hold when using different explanation methods during testing, in this experiment, we use the same model trained with Gradient × Input (thus the classification robustness is the same for different testing phases), but change two different explanation methods (Gradient and Guide Propagation) in the testing phase. The results on CIFAR10 are shown in Fig. 5, where the detailed value of this experiment can be found in Appendix Table 9. While with the same classification robustness (as shown in Table 2, under adversarial accuracy in CIFAR10), there is a huge difference between $\text{SEP}_{pos}$ and $\text{SEP}_{neg}$ w.r.t the explanation losses (both at the start and the end). This indicates even with different explanation methods in the testing phase, the explanation robustness still does not show strong correlations with adversarial robustness.

## 5.5 INFLUENCE OF DIFFERENT ADVERSARIAL TRAINING METHODS

All previous experiments utilized MAT (Madry et al., 2017) as the default adversarial training method. To assess the generalizability of our approach across different adversarial training methods, we employed TRADES (Zhang et al., 2019) in this experiment. Results in Fig. 6 (details in

Appendix Table 10) indicate that our SEP method impacts explanation robustness without altering classification robustness, suggesting a weak correlation between the two robustnesses.

## 5.6 PARAMETER SENSITIVITY ANALYSIS

In this section, we examine how different regularization weights $\lambda$ affect the results (more results in the Appendix). We trained ConvNet networks on CIFAR10 with various $\lambda$ values. Testing results are presented in Table 4. We observe that the choice of $\lambda$ influences both the exploration rate at start and end. When $\lambda$ is greater than $10^4$ or less than $-3 * 10^3$, the explanation loss changes intensely.

Table 4: The evaluation of the ConvNet trained on CIFAR10 under different $\lambda$ conditions reveals that the relationship between explanation and classification robustness is not positively correlated when an appropriate $\lambda$ is selected during model training.

| | ConvNet, CIFAR10 | | | |
|---|---|---|---|---|
| $\lambda$ | Expl at Start $(10^{-7})$ | Expl at End | Clean Acc (%) | Adv Acc (%) |
| 0 (MAT) | 16.913 | 6.206 | 64.85 | 35.11 |
| $5 * 10^4$ | 3.565 | 1.269 | 64.94 | 35.25 |
| $10^4$ | 15.436 | 5.870 | 64.39 | 35.18 |
| $10^1$ | 17.646 | 6.819 | 64.45 | 35.02 |
| $-10^2$ | 17.820 | 6.934 | 64.67 | 35.14 |
| $-3 * 10^3$ | 19.002 | 7.590 | 64.56 | 34.86 |

Table 5: Performance of different explanation methods (Gradient and Guide Propagation) in the training phase is evaluated w.r.t. explanation loss at start, at end, and adversarial accuracy on CIFAR10. Higher explanation loss at end indicates better explanation robustness, while higher adversarial accuracy denotes better classification robustness. The **best** and worst performances in explanation robustness and classification robustness are highlighted. Under various explanation methods, $\text{SEP}_{pos}$ shows a lower explanation loss compared to $\text{SEP}_{neg}$, with similar adversarial accuracy.

| | ConvNet | | | | ResNet18 | | | |
|---|---|---|---|---|---|---|---|---|
| | | | | Gradient | | | | |
| Method | Expl at Start $(10^{-7})$ | Expl at End | Clean Acc (%) | Adv Acc (%) | Expl at Start | Expl at End | Clean Acc (%) | Adv Acc (%) |
| Normal | 7.977 | 4.591 | 79.08 | 0.00 | 11.310 | 4.671 | 81.32 | 0.00 |
| MAT | 13.810 | 8.705 | 64.85 | **35.11** | 26.899 | 18.215 | 67.22 | 29.09 |
| $\text{SEP}_{pos}$ | 0.876 | 0.503 | 52.89 | 29.68 | 11.317 | 6.604 | 66.76 | **37.69** |
| $\text{SEP}_{neg}$ | 13.964 | **9.290** | 53.23 | 29.56 | 8282.990 | **7236.182** | 49.38 | 32.28 |
| | | | | Guide Propagation | | | | |
| Method | Expl at Start $(10^{-7})$ | Expl at End | Clean Acc (%) | Adv Acc | Expl at Start | Expl at End | Clean Acc (%) | Adv Acc (%) |
| Normal | 8.075 | 4.639 | 79.08 | 0.00 | 11.515 | 4.736 | 81.32 | 0.00 |
| MAT | 14.012 | 8.813 | 64.85 | 35.11 | 27.012 | 18.311 | 67.22 | 29.09 |
| $\text{SEP}_{pos}$ | 1.023 | 0.506 | 60.27 | 33.57 | 12.004 | 7.593 | 67.16 | 30.64 |
| $\text{SEP}_{neg}$ | 14.643 | **9.110** | 59.74 | **33.78** | 27.422 | **18.940** | 66.48 | **30.72** |

## 6 CONCLUSION

In summary, our study challenges the previous conclusion that explanation robustness and classification robustness are strongly correlated. Using TRADES (Zhang et al., 2019) to control explanation robustness by adjusting classification robustness, we found that increasing explanation robustness does not necessarily lead to a flatter input loss landscape for explanation loss. This contrasts with the observation that enhancing classification robustness results in a flatter input loss landscape for classification robustness. We introduce a novel algorithm to flatten the input loss landscape for explanation loss, addressing this contradiction. Our results show that our algorithm effectively improves explanation robustness without changing classification robustness, indicating a potential lack of strong correlation between the two. Our results reveal the importance of considering and optimizing both aspects separately to ensure the overall reliability and trustworthiness of AI systems in sensitive areas such as healthcare. For our future works, we hope to dive into two different robustness to understand why adversarial training can increase explanation robustness and what might be the inner difference between two robustness to understand more about the inner mechanism of adversarial attacks and explanations.

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

Table 6: Test results of models trained by Wide ResNet network and MobileNet network on various data sets according to four training methods. The results presented indicate that the performance of models trained using the Wide ResNet network and MobileNet network on different datasets suggests that there is no positive correlation between the model's explanation robustness and classification robustness achieved through the $SEP_{pos}$ and $SEP_{neg}$ training methods, as compared to the MAT training method.

| | Wide ResNet | | | | MobileNet | | | |
|---|---|---|---|---|---|---|---|---|
| | | | MNIST | | | | | |
| Method | Expl at start(1e-7) | Expl at end | Clean Acc(%) | Adv Acc | Expl at start | Expl at end | Clean Acc | Adv Acc |
| Normal | 267.050 | 206.194 | 99.58 | 0.00 | 287.061 | 188.700 | 99.08 | 0.02 |
| MAT | 842.648 | 736.839 | 98.92 | **82.82** | 4328.176 | 3356.135 | 98.29 | 94.19 |
| SEP_pos | 109.383 | 99.891 | 99.01 | 82.77 | 319.629 | 273.256 | 98.36 | **94.25** |
| SEP_neg | 937.845 | **744.698** | 98.87 | 82.71 | 8134.157 | **4454.656** | 98.33 | 94.23 |
| | | | FMNIST | | | | | |
| Method | Expl at start(1e-7) | Expl at end | Clean Acc(%) | Adv Acc | Expl at start | Expl at end | Clean Acc | Adv Acc |
| Normal | 120.037 | 69.593 | 92.79 | 0.00 | 180.159 | 103.941 | 91.93 | 0 |
| MAT | 328.817 | 257.523 | 78.10 | **68.26** | 4470.448 | 3571.210 | 68.72 | 57.19 |
| SEP_pos | 109.996 | 74.324 | 77.69 | 67.79 | 236.547 | 172.200 | 65.11 | 57.42 |
| SEP_neg | 398.006 | **304.927** | 78.21 | 68.05 | 6032.190 | **4809.288** | 66.86 | **58.16** |
| | | | CIFAR10 | | | | | |
| Method | Expl at start(1e-7) | Expl at end | Clean Acc(%) | Adv Acc | Expl at start | Expl at end | Clean Acc | Adv Acc |
| Normal | 17.920 | 8.029 | 85.47 | 0.16 | 14.797 | 6.551 | 77.48 | 0 |
| MAT | 41.513 | 27.136 | 60.01 | 24.22 | 21.502 | 13.223 | 51.51 | **23.81** |
| SEP_pos | 26.343 | 16.217 | 59.87 | 24.89 | 14.756 | 7.907 | 49.91 | 23.27 |
| SEP_neg | 43.278 | **27.575** | 60.15 | 25.08 | 26.811 | **16.420** | 35.43 | 15.30 |
| | | | CIFAR100 | | | | | |
| Method | Expl at start(1e-7) | Expl at end | Clean Acc(%) | Adv Acc | Expl at start | Expl at end | Clean Acc | Adv Acc |
| Normal | 13.677 | 5.606 | 59.13 | 0 | 17.015 | 9.351 | 43.91 | 0 |
| MAT | 30.027 | 18.389 | 36.69 | **16.12** | 20.054 | 10.836 | 21.19 | 8.64 |
| SEP_pos | 22.046 | 13.704 | 33.88 | 13.19 | 15.234 | 8.510 | 21.82 | **10.05** |
| SEP_neg | 31.889 | **20.045** | 35.74 | 15.55 | 21.544 | **13.843** | 21.35 | 7.88 |

## A   CODE AND DATA

This is our open source code link: open source code.

We conduct model training on five publicly available datasets for experiments:

- **CIFAR10** Krizhevsky et al. (2009): consisting of 60k $32 \times 32$ color images in 10 classes including 50k training and 10k test images.

- **CIFAR100** Krizhevsky et al. (2009): containing the same images as CIFAR10 but has a more refined label with 100 categories.

- **MNIST** LeCun et al. (1989): containing 60k training samples and 10k test samples from 10 digit classes. Each digit is a $28 \times 28$ grayscale image.

- **Fashion MNIST** Xiao et al. (2017): consisting of 60k training samples and $10k$ test samples from 10 classes. Each sample is a $28 \times 28$ grayscale image in a clothes category.

- **TinyImageNet** Le & Yang (2015): it is a subset of ImageNet Deng et al. (2009) with 64x64 pixels and 200 categories

## B   MORE VISUALIZATION RESULTS

Firstly, we visualize the input loss landscape w.r.t explanation loss using a normal trained model and model trained with Madry adversarial training in Fig. 7. The results show that increasing the explanation robustness does not flatten the input loss landscape. Besides, we also visualize more saliency maps with more explanation methods with images from different clusters in Fig. 8. They all prove that we can choose the most representative saliency maps.

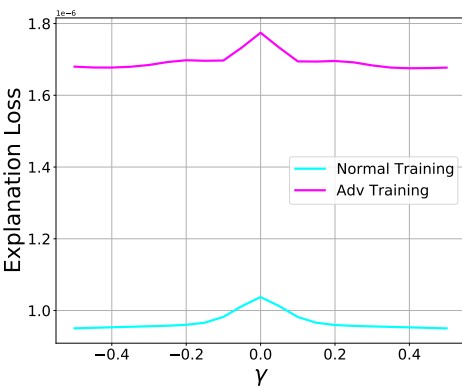

Figure 7: Comparison of input loss landscape w.r.t explanation loss with adversarial training and normal training. The results show that there is no obvious difference in input loss landscape.

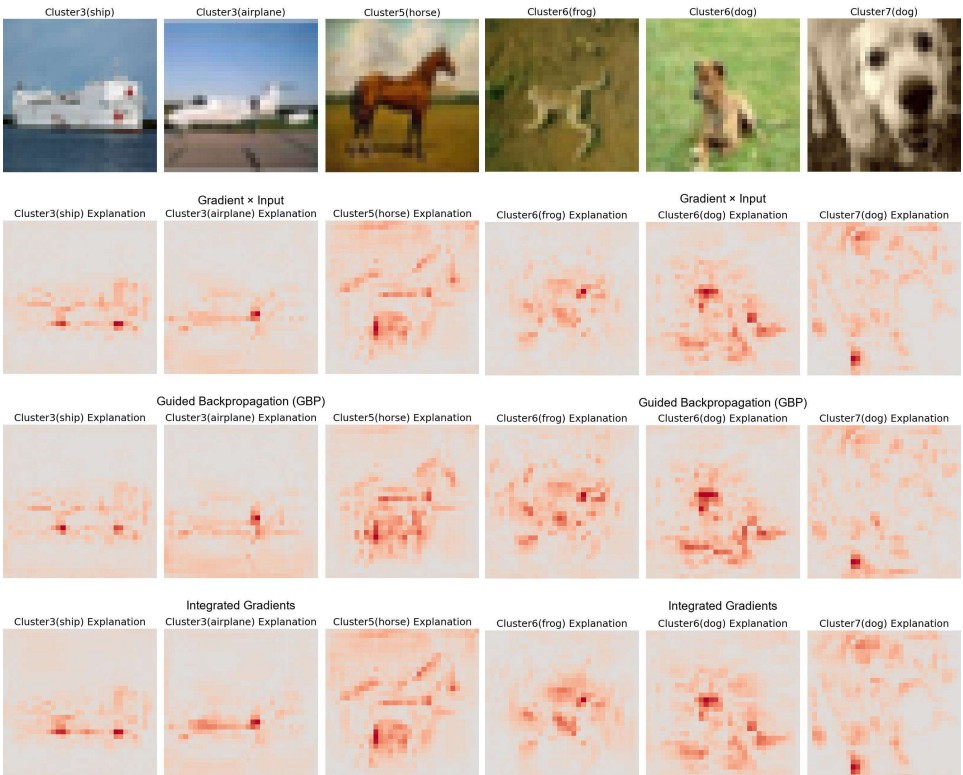

Figure 8: Explanation of images from different clusters. The results show that images from the same cluster that even have different labels still have similar saliency maps on various explanation methods. Besides, images with the same label from different clusters still have different explanations. These results show that our method can sample the most representative subset of explanations.

## C    EXPLANATION LOSS FOR INTRA-CLUSTER AND INTER-CLUSTER PAIRS

In the paper, we use Fig. 2&.8 to show the images within the same cluster share similar explanations qualitatively. Here, Table 7 shows the quantitative results of *explanation loss at start* for the intra-cluster and inter-cluster on ResNet18. We will add more results and demonstrations in the final version.

| ResNet18 | Expl at start (1e-7) |
|---|---|
| Intra-cluster | 13.726 |
| Inter-cluster | 15.437 |

Table 7: Explanation loss at start of intra and inter clusters. The smaller explanation loss in the intra-cluster shows that images in the same cluster have similar explanations.

## D    DETAILED HYPERPARAMETER

In this section, we provide the detailed hyperparameter for our CIFAR10 dataset in Table 8.

Table 8: Comparison of explanation loss for intra-cluster sample and inter-cluster sample on CI-FAR10. The results show that our cluster method indeed the cluster images with similar explanations.

| Models | Learning Rate | $\lambda$ |
|---|---|---|
| SEP_pos | | |
| ConvNet | 0.01 | 5e4 |
| ResNet18 | 0.001 | 5e4 |
| Wide ResNet | 0.001 | 5e4 |
| MobileNet | 0.01 | 5e4 |
| SEP_neg | | |
| ConvNet | 0.01 | -3e3 |
| ResNet18 | 0.001 | -1.9e3 |
| Wide ResNet | 0.001 | -1.9e3 |
| MobileNet | 0.01 | -1.25e3 |

## E    MORE EXPERIMENTAL RESULTS

### E.1    EXPERIMENTS ON W-RESNET AND MOBILENET

We list the main results using Gradient X Inputs as training and testing explanation methods for W-ResNet and MobileNetV2 in Table 6. We have the following observations:

- Once again, the adversarial accuracy for MAT, $SEP_{pos}$, and $SEP_{neg}$ is similar in most scenarios for W-ResNet and MobileNet, while $SEP_{pos}$ always has a smaller explanation loss compared with MAT, and $SEP_{neg}$ always has a larger explanation loss compared with MAT. These results show that influencing explanation robustness does not necessarily change classification robustness.
- For W-ResNet and MobileNet, the adversarial accuracy for CIFAR100 fluctuates. For MobileNet and CIFAR100, compared with MAT, $SEP_{pos}$ increases classification robustness while $SEP_{neg}$ decreases it. However, this observation also indicates that the positive correlation between explanation robustness and classification robustness might not be true since $SEP_{pos}$ decreases explanation robustness while increasing classification robustness.

### E.2    DETAILED VALUES FOR TRANSFERABLITY EXPERIMENTS

The detailed values for Transferablity experiments can be found in Table 9 and the detailed values for experiments using TRADES for our method can be found in Table 10. The analysis of these results can be found in the main paper.

Table 9: Test results for transferability of explanation robustness. Models are trained with Gradient x Input and tested on different explanation methods.All models are trained on CIFAR10. Even if the interpretation methods during training and testing are different, comparing the training results of our proposed method with the AT training method of the corresponding configuration in Table 2, we can still draw our previous conclusions, which also shows that our conclusions are transferable.

| | ConvNet | | ResNet18 | |
|---|---|---|---|---|
| | Train:Gradient X Input, Test:Gradient | | | |
| Method | Expl at start(1e-7) | Expl at end | Expl at start(1e-7) | Expl at end |
| $SEP_{pos}$ | 3.054 | 1.901 | 9.555 | 5.903 |
| $SEP_{neg}$ | 15.093 | 9.513 | 55.526 | 33.176 |
| | Train:Gradient X Input, Test:Integrated_Grad | | | |
| Method | Expl at start(1e-7) | Expl at end | Expl at start | Expl at end |
| $SEP_{pos}$ | 3.767 | 2.404 | 9.209 | 6.720 |
| $SEP_{neg}$ | 17.066 | 10.923 | 58.730 | 38.433 |

Table 10: The test results of the model trained using the TRADE training method, combined with our approach. The findings indicate that when we apply our method to TRADE, an alternative adversarial training method distinct from MAT, we can still deduce that classification robustness and interpretation robustness are not inherently interconnected.

| | ConvNet | | | |
|---|---|---|---|---|
| | CIFAR10, TRADE Weight:5 | | | |
| Method | Expl at start(1e-7) | Expl at end | Clean Acc(%) | Adv Acc(%) |
| TRADE | 18.278 | 11.470 | 64.5 | 33.98 |
| TRADE + SEP_pos | 3.878 | 2.285 | 63.84 | 33.85 |
| TRADE + SEP_neg | 19.781 | 12.424 | 64.37 | 34.07 |
| | ConvNet | | | |
| | CIFAR10, TRADE Weight:1 | | | |
| Method | Expl at start(1e-7) | Expl at end | Clean Acc(%) | Adv Acc(%) |
| TRADE | 17.271 | 10.965 | 72.63 | 28.31 |
| TRADE + SEP_pos | 4.089 | 2.296 | 72.41 | 28.20 |
| TRADE + SEP_neg | 18.504 | 11.662 | 72.90 | 28.34 |
| | ResNet18 | | | |
| | CIFAR10, TRADE Weight:5 | | | |
| Method | Expl at start(1e-7) | Expl at end | Clean Acc(%) | Adv Acc(%) |
| TRADE | 18.278 | 11.469 | 64.50 | 33.98 |
| TRADE + SEP_pos | 12.232 | 7.527 | 63.49 | 34.93 |
| TRADE + SEP_neg | 22.571 | 14.881 | 63.42 | 33.30 |

### E.3 EXPERIMENTS ON IMAGENET

Here, we present our experiments on ImageNet with ResNet18 in Table 11. We can find that the conclusion of ImageNet experiments is the same as the main paper: Increasing or decreasing explanation robustness will not necessarily influence the classification robustness.

| | Expl at start (1e-7) | Expl at end (1e-7) | Adv Acc (%) |
|---|---|---|---|
| Normal | 114.70 | 63.52 | 0.00 |
| AT | 1281.71 | 742.43 | 19.36 |
| $SEP_{pos}$ | 287.64 | 156.16 | 17.63 |
| $SEP_{neg}$ | 1427.33 | 905.25 | 17.44 |

Table 11: Experiments for ImageNet on ResNet18. The results are aligned with the conclusion made in the main paper.

### E.4 MORE EXPERIMENTS ON DIFFERENT EXPLANATION METHODS

We provide more results for FashionMnist and TinyImageNet on ConvNet and ResNet using Guide Propagation as the explanation method in Table 15. We also provide the experimental results for DeepLift and Integrated Gradients in Table 12. The results show a similar conclusion in the main text, where it is possible to influence the explanation robustness without changing adversarial robustness, which demonstrates that our conclusion works in general for different explanation methods.

Table 12: Performance of using DeepLift and Intergrated Gradients as explanation methods with ConvNet. Higher explanation loss at end indicates better explanation robustness, while higher adversarial accuracy denotes better classification robustness. The **best** and worst performances in explanation robustness and classification robustness are highlighted. Under various explanation methods, $SEP_{pos}$ shows a lower explanation loss compared to $SEP_{neg}$, with similar adversarial accuracy.

| | DeepLift | | | | Integrated Gradients | | | |
|---|---|---|---|---|---|---|---|---|
| | | | | MNIST | | | | |
| Method | Expl at Start $(10^{-7})$ | Expl at End | Clean Acc(%) | Adv Acc(%) | Expl at Start | Expl at End | Clean Acc(%) | Adv Acc (%) |
| MAT | 369.153 | 294.053 | 99.00 | 89.92 | 239.650 | 224.745 | 99.00 | 89.92 |
| $SEP_{pos}$ | 82.959 | 57.408 | 98.93 | 95.97 | 76.284 | 50.603 | 98.42 | **93.19** |
| $SEP_{neg}$ | 1101.038 | **896.157** | 98.97 | **96.16** | 778.663 | **534.113** | 98.68 | 92.76 |
| | | | | FMNIST | | | | |
| Method | Expl at Start $(10^{-7})$ | Expl at End | Clean Acc(%) | Adv Acc(%) | Expl at Start | Expl at End | Clean Acc(%) | Adv Acc (%) |
| MAT | 386.377 | 274.130 | 62.85 | **73.98** | 237.727 | 234.109 | 62.85 | **73.98** |
| $SEP_{pos}$ | 33.824 | 21.429 | 60.75 | 65.52 | 21.425 | 16.048 | 60.85 | 72.05 |
| $SEP_{neg}$ | 4739.769 | **3153.331** | 60.62 | 70.05 | 3624.231 | **2748.976** | 62.92 | 70.36 |
| | | | | CIFAR10 | | | | |
| Method | Expl at Start $(10^{-7})$ | Expl at End | Clean Acc(%) | Adv Acc(%) | Expl at Start | Expl at End | Clean Acc(%) | Adv Acc(%) |
| MAT | 18.137 | 11.562 | 64.85 | 35.11 | 16.887 | 14.007 | 64.85 | 35.11 |
| $SEP_{pos}$ | 2.593 | 1.273 | 65.76 | **35.38** | 3.696 | 2.523 | 60.19 | 32.33 |
| $SEP_{neg}$ | 21.329 | **13.819** | 64.20 | 34.91 | 19.568 | **14.803** | 60.86 | 32.43 |
| | | | | CIFAR100 | | | | |
| Method | Expl at Start $(10^{-7})$ | Expl at End | Clean Acc(%) | Adv Acc(%) | Expl at Start | Expl at End | Clean Acc(%) | Adv Acc(%) |
| MAT | 19.683 | 12.766 | 36.40 | 17.35 | 16.754 | 10.779 | 36.40 | **17.35** |
| $SEP_{pos}$ | 14.208 | 9.201 | 39.10 | 18.23 | 5.320 | 3.218 | 34.73 | 16.74 |
| $SEP_{neg}$ | 20.389 | **13.694** | 39.78 | **18.32** | 17.011 | **11.356** | 35.10 | 16.60 |

## E.5 MORE PARAMETER SENSITIVITY STUDIES

**Training Epochs** We conducted experiments on the ConvNet network using the CIFAR10 dataset to show that our chosen training epoch is reasonable. The results, as presented in Table 13, indicate that the model's performance undergoes only marginal changes after 25 rounds for ConvNet, despite the epoch count continuing to increase. Choosing 25 epochs does not hurt the reliability of our argument. Besides, the results also support our conclusion. With the increase of training epochs, the classification robustness still increases while the explanation robustness actually decreases.

Table 13: The test results of ConvNet network at different training epochs on the CIFAR10 data set.The findings indicate that as we increase the number of training epochs from 25, there is only marginal improvement in the model's performance for ConvNet. Therefore, we have decided to select 25 epochs as the final number of training epochs for all our models. This choice will not impact our final conclusions, while also allowing for faster training speed.

| ConvNet, CIFAR10 | | | | |
|---|---|---|---|---|
| Training Epoch | Expl at start(1e-7) | Expl at end | Clean Acc(%) | Adv Acc (%) |
| 25 | 4.388 | 1.605 | 64.94 | 35.25 |
| 50 | 3.885 | 1.431 | 65.69 | 35.94 |
| 75 | 3.671 | 1.378 | 66.33 | 36.27 |
| 100 | 3.557 | 1.339 | 66.74 | 36.50 |

Table 14: The results of ResNet18 with different training epochs on CIFAR10. The results show that with increasing training epochs, the accuracy of ResNet18 on CIFAR10 keeps increasing while our conclusion remains the same.

| Training Epochs 70 | | | | |
|---|---|---|---|---|
| Method | Expl at start(1e-7) | Expl at end | Clean Acc(%) | Adv Acc(%) |
| MAT | 33.620 | 24.310 | 68.16 | 39.43 |
| MAT + SEP_pos | 11.594 | 9.604 | 68.67 | 39.58 |
| Training Epoch 100 | | | | |
| Method | Expl at start(1e-7) | Expl at end | Clean Acc(%) | Adv Acc(%) |
| MAT | 34.706 | 25.179 | 71.31 | 40.55 |
| MAT + SEP_pos | 9.682 | 8.294 | 71.18 | 40.43 |

Table 15: Performance of using Guide Propagation in the training phase with Fashion-MNIST and TinyImageNet. Higher explanation loss at end indicates better explanation robustness, while higher adversarial accuracy denotes better classification robustness. The **best** and worst performances in explanation robustness and classification robustness are highlighted. Under various explanation methods, $SEP_{pos}$ shows a lower explanation loss compared to $SEP_{neg}$, with similar adversarial accuracy.

| | ConvNet | | | | ResNet18 | | | |
|---|---|---|---|---|---|---|---|---|
| | | | FMNIST | | | | | |
| Method | Expl at Start $(10^{-7})$ | Expl at End | Clean Acc (%) | Adv Acc (%) | Expl at Start | Expl at End | Clean Acc (%) | Adv Acc (%) |
| Normal | 30.932 | 18.451 | 92.79 | 0.00 | 66.131 | 29.110 | 91.57 | 0.00 |
| MAT | 97.726 | 72.402 | 62.85 | 73.98 | 608.486 | 467.815 | 79.22 | 67.10 |
| $SEP_{pos}$ | 48.368 | 34.672 | 78.46 | 67.28 | 97.703 | 78.354 | 77.55 | 62.09 |
| $SEP_{neg}$ | 542.540 | 425.948 | 65.07 | 77.17 | 4219.351 | 3839.408 | 80.11 | 72.21 |
| | | | TinyImageNet | | | | | |
| Method | Expl at Start $(10^{-7})$ | Expl at End | Clean Acc (%) | Adv Acc | Expl at Start | Expl at End | Clean Acc (%) | Adv Acc (%) |
| Normal | 0.559 | 0.281 | 28.71 | 0.00 | 0.617 | 0.216 | 28.34 | 0.00 |
| MAT | 1.356 | 0.787 | 25.13 | 9.55 | 2.577 | 1.411 | 26.33 | 10.81 |
| $SEP_{pos}$ | 0.983 | 0.625 | 25.16 | 5.97 | 1.767 | 1.226 | 28 68 | 11.47 |
| $SEP_{neg}$ | 1.566 | 0.977 | 24.89 | 4.99 | 3.403 | 1.761 | 26.79 | 11.23 |

