# OpenReview forum: "Are Classification Robustness and Explanation Robustness Really Strongly Correlated? An Analysis Through Input Loss Landscape"
_ICLR.cc/2025/Conference — ICLR 2025 Conference Withdrawn Submission_

### Official Review · Reviewer_HLgC · 2024-10-30

**Soundness:** 2
**Presentation:** 2
**Contribution:** 2
**Rating:** 3
**Confidence:** 2

**Summary:**

The paper looks at the relationship between classification robustness and explanation robustness. Previous works have claimed that improving explanation robustness leads to improved classification robustness. The authors empirically show that models trained adversarial are not robust to perturbations effecting the explanation robustness, and propose a regularisation term to enforce that images with additive Gaussian noise achieve similar explanations. The proposed method achieves similar classification robustness, but improved explanation robustness, and is demonstrated in a number of settings.

**Strengths:**

- The topic is very interesting, looking at robustness to perturbations affecting classification performance and explanation similarity.
- The breadth of experiments performed is impressive.
- The related work section goes over a lot of methods for both adversarial training and the robustness of explanations.

**Weaknesses:**

- The paper is difficult to follow, it could benefit from trimming down the content to simplify the message. The content is interesting, but it is presented in a way that is difficult to digest.
- Some notation is not defined until Section 4 but is used in previous sections.
- In Appendix C, the intra-cluster and inter-cluster explanation loss seems very similar, and without reporting standard deviation it is difficult to assess the performance. The appendix could benefit from more visual examples within clusters like Figure 3.
- In Figure 4, the caption should use $\alpha$ instead of trade weight.
- An overview of the explanation methods used in 5.1 would be good, to give the reader intuition on why the performance is different.
- Some of the claims in the paper are contradictory, and it is not clear the difference. In Section 2 line 187 it states 'increasing explanation robustness can benefit classification robustness', then in 5.2 line 415 'there is no inherent relationship between explanation robustness and classification robustness'. If the authors work disagrees with existing literature, then it should be framed in such a that it is clear to the reader that the results disagrees with other literature.
- Line 97 needs citations for the claim made.
- Multiple definitions of 'Adversarial Training (AT)' can be found, particularly in line 135 and 136.

**Questions:**

- In Figure 1, are the images classified as the same? There are some details lacking in the caption.
- For $SEP_{pos}$ and $SEP_{neg}$, why were the values chosen to be 50,000 and -3000 respectively. The negative value is much less and it is not clear why.
- Do the authors believe that with a different training criteria, that the explanations should still be similar? The clustering is performed using one network, and these representative samples are used to evaluate all other networks. Taking an explanation from a pre-trained ResNet18 as the ground truth for evaluating all models to feels like a very strong assumption.
- Visually, can you see a difference between the explanations in each cluster?
- How important is the clustering approach to select certain types of explanations? Another approach could be for each image you randomly select a subset of other images to perform the adversarial attack on the explanation. Over each training epoch, each image would see a different subset.

---

### Official Review · Reviewer_vkBN · 2024-11-02

**Soundness:** 2
**Presentation:** 2
**Contribution:** 2
**Rating:** 3
**Confidence:** 4

**Summary:**

The paper challenges the common belief that classification robustness and explanation robustness in image classification systems are inherently linked. It reveals that enhancing explanation robustness does not necessarily flatten the input loss landscape with respect to explanation loss. Additionally, it finds that while adjusting the loss landscape with respect to explanation loss can affect the robustness of the explanations, it does not impact classification robustness.

**Strengths:**

1.	Intriguing Research Question: The study is addressing the compelling question of whether classification robustness and explanation robustness are inherently correlated in image classification systems.
2.	The paper provides ample examples and detailed experimental results.

**Weaknesses:**

**1. I held reservation on conclusions drawn on correlation between explanation robustness and classification robustness.**

The paper asserts that "explanation robustness may not be strongly correlated to classification robustness" based on experiments using a training method that flattens the input loss landscape with respect to explanation loss. This training method decreases the explanation robustness but not change the classification robustness measured by adversarial accuracy.

To understand the relationship between explanation robustness and classification robustness, one could explore whether methods aimed at enhancing explanation robustness also contribute to improvements in adversarial accuracy. Various strategies have been proposed to improve explanation robustness against adversarial attacks, as detailed in Table 3 of the referenced document (https://arxiv.org/pdf/2306.06123).

Based on the findings presented in the paper, the following conclusions can be drawn:

(a) Flattening the loss landscape with respect to explanation loss does not effectively enhance explanation robustness;

(b) This flattening approach does not significantly alter the classification robustness, as measured by adversarial accuracy.

However, combining observations (a) and (b) does not necessarily lead to the conclusion that explanation robustness is not strongly correlated with classification robustness.

**2. The current version of the paper is not well written with quite a lot of unclarity and organization issues.**

**Organizational Issues:**

a. The conclusion presented in Lines 41-42 is what the paper wants to challenge, yet it is positioned as if it were the primary conclusion of the paper. This could confuse readers about the paper's main message.

b. In the beginning of Section 3, the paper states, "In the previous section, we observe a strange situation where increasing the explanation robustness does not flatten the input loss landscape with respect to explanation robustness." However, this issue is not adequately discussed until Section 4, requiring readers to refer back to subsequent sections for a full understanding. Similarly, Line 93 of the introduction references Figure 4, forcing readers to navigate back and forth through the document repeatedly.

**Missing Details:**

a. The explanation methods used in Figure 1 and Figure 3, and in Table 1, are not specified.

b. Line 96-97 claims that “previous works have proven that increasing classification robustness can also increase explanation robustness”, yet no references are provided to support this assertion.

c. The term "victim image" is used without definition. While referenced (Tamam et al., 2022; Dombrowski et al., 2019), these sources do not clearly define what constitutes a "victim image," which may lead to ambiguity in understanding the term’s application.

d. In Line 281-283 and Table 1, it is necessary to clarify that the "white-box attack method (Dombrowski et al., 2019)" refers to the attack method specifically targeting explanations. Some introduction to the attack method in necessary.

**3.	The experimental results as presented may not sufficiently support the claim on a broader scale.**

a. Currently, all evaluation methods discussed are gradient-based. It remains to be seen whether these findings are applicable to model-agnostic methods such as LIME [1], or to perturbation-based explanation methods like RISE [2] and Score-CAM [3]. The paper does not clarify whether the results extend to these types of methods.

b. The evaluations could also be expanded to include Vision Transformers (ViT) and employ explanation methods specifically designed for these models, as referenced in [5][6][7].

[1] " Why should i trust you?" Explaining the predictions of any classifier.

[2] RISE: Randomized Input Sampling for Explanation of Black-box Models.

[3] Score-CAM: Score-Weighted Visual Explanations for Convolutional Neural Networks.

[4] Transformer interpretability beyond attention visualization.

[5] Generic Attention-model Explainability for Interpreting Bi-Modal and Encoder-Decoder Transformers.

[6] Vit-cx: Causal explanation of vision transformers.

**Questions:**

See weaknesses.

---

### Official Review · Reviewer_SmMi · 2024-11-02

**Soundness:** 3
**Presentation:** 1
**Contribution:** 2
**Rating:** 5
**Confidence:** 3

**Summary:**

The paper considers the link between classification robustness and explanation robustness and provides empirical evidence that, unlike for classification robustness, explanation robustness does not lead to a flatten loss landscape. The authors propose a method that directly incorporates the flattening objective in the training and illustrate that this does not affect the classification robustness, while impacting the explanation robustness, leading to the conclusion that classification robustness and explanation robustness are not strongly correlated. Empirical results are provided across 6 datasets and various backbones and explanation methods.

**Strengths:**

Understanding the links between classification and explanation robustness is an interesting and relevant problem that can inform the design of more robust models.

The evaluation considers a range of backbones and explanation approaches and results generally support the claim that it is possible to modify the explanation robustness without affecting the classification robustness.

**Weaknesses:**

Lack of discussion of insights/findings:

Right now, the work illustrates the case where increasing explanation robustness does not necessarily increase classification robustness, which is achieved by encouraging flat or sharper loss surfaces. However, the authors provide no explanation/insight into why sharper loss surfaces lead to increased explanation robustness. Further, can the authors discuss how these results relate to the theoretical results on this topic by (Huang et al, 2023)?

Presentation:
- The current presentation can be significantly improved. For instance, there seems to have been some re-shuffeling of the sections without modification of the flow. Right now chapter 3 (Methods) refers to the observations from the “previous section”, which however is not presented until Chapter 4. In Chapter 4, on the other hand, there are references to the method that will be presented in the next chapter, but was already presented in Chapter 3.
- Similarly, in Chapter 3, the chosen training framework is stated as Madry, while in Chapter 4 it is stated as TRADES instead of Madry. - Appendix A is missing the sixth dataset.
- The caption of Table 8 does not seem to match the Table.
- The notation for the gradient of the loss function in Algorithm 1 does not align with Eq. 2

Experimental evaluation:
- In Table 4, the explanation robustness is provided for different lambda values and only seems to be increasing as the lambda becomes increasingly negative. Does this trend continue as lambda decreases?
- Results are reported over single runs and no analysis of the variability/significance are performed.
- Why are ImageNet results compared to Adversarial Training (AT) and not MAT in Appendix E.3?
- Hyperparameter choices are only provided for CIFAR-10 and not the remaining datasets/models.

**Questions:**

The reviewers main concerns are on the lack of discussion of the insights and findings as well as the experimental evaluation.

In the rebuttal, could the authors provide insights into why sharper loss surfaces lead to increased explanation robustness and relate their findings to the theoretical results by (Huang et al, 2023)?

For the experimental evaluation, the reviewers main concerns are related to the behavior as lambda decreases as well as on the variability of the results over multiple runs. Could the authors provide a robustness analysis?

---

### Official Review · Reviewer_mMVd · 2024-11-03

**Soundness:** 1
**Presentation:** 3
**Contribution:** 2
**Rating:** 3
**Confidence:** 4

**Summary:**

This paper examines the concept of explanation robustness through the lens of classification robustness in the context of targeted adversarial attacks. While classification robustness traditionally measures a model’s accuracy when facing adversarial attacks, the authors define explanation robustness as the deviation or error between the “victim” explanation under adversarial attack on the input and the targeted explanation (L79). Although the paper suggests that improving one type of robustness is commonly believed to inherently enhance the other, this assumption could benefit from further investigation. In fact, the study’s results indicate that classification robustness and explanation robustness are not inherently correlated.

**Strengths:**

In this paper, the authors take an intriguing and bold approach by applying the connection between classification robustness and a flattened loss surface to the concept of explanation robustness. Their method of linking this idea with adversarial training is seamless and demonstrates a thoughtful integration of ideas. Additionally, the design and execution of experiments to objectively validate their claims are commendable, as they provide a strong empirical foundation to support the authors' arguments. This structured and rigorous approach enhances the paper’s credibility and showcases a well-rounded effort to explore and substantiate their innovative perspective on robustness.

**Weaknesses:**

The paper claims to challenge a 'conventional belief in a strong and inherent correlation between classification robustness and explanation robustness. However, this assumption, the two robustness types are inherently linked and that improving one will necessarily enhance the other, is neither universally accepted nor rigorously established. Rather, it seems to be an implicit assumption rather than a widely confirmed principle. Although some prior works, such as Boopathy et al. (2020), suggest that bolstering interpretability may indirectly support classification robustness, this relationship is contextual and does not imply a direct or reciprocal correlation.

Your results confirm that classification robustness and explanation robustness address fundamentally different aspects of model behavior. Classification robustness pertains to the model’s predictive stability under adversarial examples, while explanation robustness concerns the consistency of interpretive outputs. For instance, consider a model with a feature extractor trained using a self-supervised learning method first, where only the classification head is trained with labeled data afterwards. In such cases, these two types of robustness do not inherently align, making it unreliable to assume a straightforward correlation between them without considering specific contexts and dependencies.

**Questions:**

1. The paper suggests that classification robustness and explanation robustness are inherently linked. Could the authors provide a stronger theoretical foundation or rationale for why this link should exist beyond the papers you cited? What specific mechanisms would explain a causal relationship between these two forms of robustness?

2.   The paper [1] introduces indiscriminate adversarial attacks on unsupervised contrastive learning, revealing vulnerabilities in SSL-trained models. Considering this, have the authors tested whether the proposed approach for explanation robustness holds under poisoning attacks in self-supervised learning frameworks? Specifically, how would a model trained with SSL and subsequently fine-tuned with a supervised classification head respond to adversarial attacks on explanation robustness? Additionally, could the approach accommodate scenarios where explanations are generated from SSL-trained feature extractors, as discussed in He et al.’s work?

[1] He, H., Zha, K., & Katabi, D. Indiscriminate Poisoning Attacks on Unsupervised Contrastive Learning. In The Eleventh International Conference on Learning Representations, 2023.

3.

---

### Note · Authors · 2024-11-15

I have read and agree with the venue's withdrawal policy on behalf of myself and my co-authors.